# Platelet-Derived Extracellular Vesicles Stimulate Migration through Partial Remodelling of the Ca^2+^ Handling Machinery in MDA-MB-231 Breast Cancer Cells

**DOI:** 10.3390/cells11193120

**Published:** 2022-10-04

**Authors:** Mauro Vismara, Sharon Negri, Francesca Scolari, Valentina Brunetti, Silvia Maria Grazia Trivigno, Pawan Faris, Luca Galgano, Teresa Soda, Roberto Berra-Romani, Ilaria Canobbio, Mauro Torti, Gianni Francesco Guidetti, Francesco Moccia

**Affiliations:** 1Laboratory of Biochemistry, Department of Biology and Biotechnology “L. Spallanzani”, University of Pavia, 27100 Pavia, Italy; 2Laboratory of General Physiology, Department of Biology and Biotechnology “L. Spallanzani”, University of Pavia, 27100 Pavia, Italy; 3Institute of Molecular Genetics, Italian National Research Council (CNR), 27100 Pavia, Italy; 4University School for Advanced Studies IUSS, 27100 Pavia, Italy; 5Department of Biology, College of Science, Salahaddin University, Erbil 44001, Kurdistan-Region of Iraq, Iraq; 6Department of Health Sciences, University of Magna Graecia, 88100 Catanzaro, Italy; 7Department of Biomedicine, School of Medicine, Benemérita Universidad Autónoma de Puebla, Puebla 74325, Mexico

**Keywords:** platelets, extracellular vesicles, breast cancer, migration, Ca^2+^ signalling, SERCA2B, InsP_3_ receptors, myosin light chain 2, p38 mitogen-activated protein kinase

## Abstract

Background: Platelets can support cancer progression via the release of microparticles and microvesicles that enhance the migratory behaviour of recipient cancer cells. We recently showed that platelet-derived extracellular vesicles (PEVs) stimulate migration and invasiveness in highly metastatic MDA-MB-231 cells by stimulating the phosphorylation of p38 MAPK and the myosin light chain 2 (MLC2). Herein, we assessed whether the pro-migratory effect of PEVs involves the remodelling of the Ca^2+^ handling machinery, which drives MDA-MB-231 cell motility. Methods: PEVs were isolated from human blood platelets, and Fura-2/AM Ca^2+^ imaging, RT-qPCR, and immunoblotting were exploited to assess their effect on intracellular Ca^2+^ dynamics and Ca^2+^-dependent migratory processes in MDA-MB-231 cells. Results: Pretreating MDA-MB-231 cells with PEVs for 24 h caused an increase in Ca^2+^ release from the endoplasmic reticulum (ER) due to the up-regulation of SERCA2B and InsP_3_R1/InsP_3_R2 mRNAs and proteins. The consequent enhancement of ER Ca^2+^ depletion led to a significant increase in store-operated Ca^2+^ entry. The larger Ca^2+^ mobilization from the ER was required to potentiate serum-induced migration by recruiting p38 MAPK and MLC2. Conclusions: PEVs stimulate migration in the highly metastatic MDA-MB-231 breast cancer cell line by inducing a partial remodelling of the Ca^2+^ handling machinery.

## 1. Introduction

Breast cancer represents the most widespread cancer in women and accounted for about 685.000 deaths in 2020. This death toll has been forecasted to rise to 7 million by 2040 [1]. Triple negative breast cancer (TNBC) is featured by the absence of the human epidermal growth factor receptor 2 (HER2), estrogen receptor, and progesterone receptor. Therefore, TNBC is barely sensitive to anti-HER2 and hormonal therapies, with poor survival rates [2]. Accordingly, TNBC presents high aggressiveness and short median time to relapse due to its ability to leave the primary tumour site and spread to distant organs, thereby causing patient death [3]. Understanding the cellular and molecular mechanisms that stimulate TNBC cells to migrate and colonize their new niches is indispensable when it comes to designing alternative treatments for TNBC patients.

Platelet-derived extracellular vesicles (PEVs) are gaining growing interest as mediators of platelet function in different physiological and pathological contexts, from haemostasis to cardiovascular diseases [4,5]. In the last two decades, PEVs have also been recognized as critical players in the complex interplay occurring between blood platelets and cancer [6]. Exosomes and plasma membrane-derived vesicles are the two main classes of PEVs released by platelets into the bloodstream [4]. Platelet exosomes (also known as small PEVs) are stored in intracellular multivesicular bodies and released during cell secretion. Conversely, membrane-derived vesicles derive from the plasma membrane and are now typically defined as medium-large PEVs, but they were previously referred to as platelet-derived microparticles (PMPs) and platelet-derived microvesicles (PMVs) [7].

Medium-large PEVs (henceforth PEVs for brevity) have been widely studied in the frame of the platelet–cancer crosstalk. Their levels in the circulation are increased in oncological patients bearing diverse types of cancer, including cutaneous malignant melanoma [8], colorectal carcinoma [9], lung cancer [10], and breast cancer [11]. Moreover, PEVs were found to mediate intercellular communication by delivering bioactive compounds, thus initiating phenotypic and functional changes in recipient cells. These observations boosted research on the role played by these PEVs in cancer aiming to unravel their contribution in the progression of the disease but also to determine their potential use as early diagnostic markers and novel drug delivery tools [12,13]. PEVs are now known to directly regulate cancer cells as well as the cellular components of the tumour-microenvironment (TME). In this context, most studies have hinted at PEVs as crucial drivers of cancer progression, although a few investigations have discussed their anti-cancer functions [6,14,15].

We showed that thrombin-induced PEVs are internalized by different breast cancer cell lines, thereby exerting cell-specific reactions [16]. In particular, the TNBC MDA-MB-231 cell line efficiently internalizes PEVs, and this event potentiates cell migration and invasiveness. In agreement with these observations, long-term (up to 24 h) exposure to PEVs stimulates specific signalling pathways, such as p38 mitogen-activated protein kinase (p38 MAPK), myosin light chain-2 (MLC2), and Rho-associated protein kinase (ROCK) [16], that drive cancer cell motility and spreading [17,18,19,20]. Nevertheless, the molecular mechanism(s) whereby a prolonged exposure to PEVs promote(s) migration in MDA-MB-231 cells is still unclear. The remodelling of the Ca^2+^ handling machinery supports several cancer hallmarks, including tissue invasion and metastasis [21,22,23,24,25]. An increase in intracellular Ca^2+^ concentration ([Ca^2+^]_i_) in MDA-MB-231 cells can be elicited by inositol-1,4,5-trisphosphate (InsP_3_) [26], which gates three subtypes of InsP_3_ receptors (InsP_3_R1, InsP_3_R2, and InsP_3_R3) to release Ca^2+^ from the endoplasmic reticulum (ER) [22]. InsP_3_-induced Ca^2+^ mobilization, in turn, causes a strong reduction in ER Ca^2+^ concentration, which leads to the activation of a Ca^2+^ entry pathway on the plasma membrane, known as store-operated Ca^2+^ entry (SOCE) [27,28,29]. In MDA-MB-231 cells, SOCE is mediated by the physical association between Stromal Interaction Molecule 1 (STIM1), which functions as a sensor of ER Ca^2+^ concentration and is activated by a drop in intraluminal Ca^2+^ levels, and Orai1, which forms the Ca^2+^-permeable channel on the plasma membrane [24,27,30,31]. The interaction between InsP_3_-induced ER Ca^2+^ mobilization and SOCE finely shapes the intracellular Ca^2+^ signals that increase the migration capacity of this highly invasive breast cancer cell line [24,26,31,32]. An increase in the expression of genes encoding for several InsP_3_R isoforms can stimulate multiple cancer hallmarks, including proliferation, migration, invasion, and apoptosis resistance [33,34]. Similarly, SOCE up-regulation, because of the over-expression of STIM and/or Orai1 proteins, can result in the activation of many pro-oncogenic signalling pathways in neoplastic cells [22,29,35]. Furthermore, the overexpression of several members of the Transient Receptor Potential (TRP) superfamily of non-selective cation channels can also support neoplastic transformation in virtually all cancer cell types [22,36].

Intriguingly, the signalling pathways recruited downstream of PEV stimulation, e.g., p38 MAPK and MLC2, can be activated following an elevation in [Ca^2+^]_i_ [16,18]. Herein, we thus sought to assess whether long-term exposure to PEVs stimulates MDA-MB-231 cell migration through the remodelling of the Ca^2+^ handling machinery. By using a variety of approaches, we showed that PEVs cause a remarkable elevation in InsP_3_-induced ER Ca^2+^ release by increasing Sarco-Endoplasmic Ca^2+^-ATPase 2B (SERCA2B) and InsP_3_R1/InsP_3_R2 transcript and protein expression. The larger depletion in ER Ca^2+^ content, in turn, leads to enhanced SOCE activation. In agreement with this observation, serum elicited larger intracellular Ca^2+^ signals and potentiated migration in MDA-MB-231 cells exposed to PEVs through the Ca^2+^-dependent recruitment of p38 MAPK and MLC2. Our evidence indicates that InsP_3_Rs rather than SOCE are required to drive MDA-MB-231 cell migration. These data shed novel light on the signalling pathways whereby PEVs can stimulate the metastatic spreading of TNBC cells and hint at the Ca^2+^ toolkit as a promising target to prevent the detrimental interaction between PEVs and these highly aggressive breast cancer cells.

## 2. Materials and Methods

### 2.1. Cancer Cell Culture

The triple negative breast cancer line, MDA-MB-231, was provided by Professor Livia Visai (Department of Molecular Medicine, University of Pavia). Cancer cells were periodically checked to verify the absence of bacterial contamination and were maintained in DMEM supplemented with 10% foetal bovine serum (FBS), 2 mM L-glutamine, 100 U/mL penicillin, and 100 μg/mL streptomycin, split every two days, and used for the experiments within 10 passages. The count of vital cells was determined by Trypan Blue staining and phase contrast microscopy analysis.

### 2.2. PEV Isolation

Human blood platelets were purified from healthy donors as recently described [16]. Upon the washing procedure, platelets were resuspended at a concentration of 3 × 10^8^/mL in HEPES buffer (10 mM HEPES, 137 mM NaCl, 2.9 mM KCl, and 12 mM NaHCO_3_, pH 7.4) supplemented with 1 mM CaCl_2_, 0.5 mM MgCl_2_ and 5.5 mM glucose. To induce the release of PEVs, platelets were stimulated with the physiological agonist thrombin (0.2 U/mL) for 30 min at 37 °C under constant stirring. Platelets were pelleted by low-speed centrifugation (750× *g*, 20 min) and the supernatant was then centrifuged at 18,500× *g* for 90 min at 10 °C to collect medium-large PEVs, which were finally resuspended in HEPES buffer. The protein content of the different preparations of PEVs was determined by BCA assay.

### 2.3. [Ca^2+^]_i_ Measurements

As described elsewhere, Ca^2+^ imaging was used to measure intracellular Ca^2+^ signals in PEV-treated cancer cells [16,37]. MDA-MB-231 cells were loaded with 4 μM Fura-2/AM (1 mM stock in DMSO) in physiological salt solution (PSS) (150 mM NaCl, 6 mM KCl, 1.5 mM CaCl_2_, 1 mM MgCl_2_, 10 mM Glucose, 10 mM HEPES, pH 7.4) for 30 min at 37 °C and 5% CO_2_. After washing in PSS, the coverslip was fixed to the bottom of a Petri dish and the cells were either left untreated or treated with 30 µg/mL thrombin-induced PEVs. The cells were observed by an upright epifluorescence Axiolab microscope (Carl Zeiss, Oberkochen, Germany) equipped with a Zeiss ×40 Achroplan objective (water-immersion, 2.0 mm working distance, 0.9 numerical aperture). The cells were excited alternately at 340 and 380 nm, and the emitted light was detected at 510 nm. Custom software, working in the LINUX environment, was used to drive the camera (Extended-ISIS Camera, Photonic Science, Millham, UK) and the filter wheel, and to measure and plot on-line the fluorescence from rectangular “regions of interest” (ROI) enclosing 20–30 single cells. [Ca^2+^]_i_ was monitored by measuring, for each ROI, the ratio of the mean fluorescence emitted at 510 nm when exciting alternatively at 340 and 380 nm [ratio (F_340_/F_380_)]. An increase in [Ca^2+^]_i_ causes an increase in the ratio [38,39]. Ratio measurements were performed and plotted on-line every 3 s. The experiments were performed at room temperature (22 °C).

The resting Ca^2+^ entry was evaluated by exploiting the Mn^2+^-quenching technique. Mn^2+^ has been shown to quench Fura-2/AM fluorescence. Since Mn^2+^ and Ca^2+^ share common entry pathways in the plasma membrane, Fura-2/AM quenching by Mn^2+^ is regarded as an index of divalent cation influx [31,40,41]. Experiments were carried out at the 360 nm wavelength, the isosbestic wavelength for Fura-2/AM, and in Ca^2+^-free medium supplemented with 0.5 mM EGTA, as previously described [40,42]. This avoids Ca^2+^ competition for Mn^2+^ entry and therefore enhances Mn^2+^ quenching. The basal [Ca^2+^]_i_ was evaluated by using the Grynkiewicz equation, as shown elsewhere [35,43].

### 2.4. Cell Migration Assay

The effect of PEVs on the migration of MDA-MB-231 cells was evaluated using Falcon cell culture inserts (8 μm pore size) positioned in a 24-well plate, as shown in [16]. Cancer cells were serum starved for 6 h and then resuspended in DMEM with the addition of 0.5% FBS. Cells were either left untreated or treated with PEVs, supplemented or not with 2-Aminoethyl diphenyl borate (2-APB; 50 µM) or YM-58483 (also known as BTP-2; 20 µM), and then transferred inside the inserts. DMEM containing 10% FBS was added to the lower chamber. After 24 h, cells that moved through the porous membrane were stained with 0.5% crystal violet and counted at 10× microscope magnification with an Olympus BX51 microscope (Olympus Corporation, Japan).

### 2.5. Real-Time Quantitative Reverse Transcription PCR (qRT-PCR)

The total RNA was isolated from PEVs as well as PEV-treated and untreated MDA-MB-231 cells using Trizol reagent (Thermo Fisher Scientific, Waltham, MA, USA) according to the manufacturer’s instructions. For RNA extraction from PEVs, 20 µg of RNA grade glycogen (Thermo Fisher Scientific) was added in the precipitation step [44]. After DNAse treatment (Turbo DNA-free™ kit, Thermo Fisher Scientific), RNA was quantified with a BioPhotometer D30 (Eppendorf, Hamburg, Germany). cDNA was synthetised from 200 ng of MDA-MB-231 RNA and 100 ng of PEV RNA using the iScript cDNA Synthesis Kit (BioRad, Hercules, CA, USA). Gene expression analyses were performed in triplicate with specific primers (Appendix A) and the SsoFast™ EvaGreen^®^ Supermix (BioRad) using a CFX Connect Real-Time System (BioRad) using the following program: an initial step at 95 °C for 30 s, 40 cycles of 5 s at 95 °C, and 5 s at 58 °C. Fluorescence measurements were taken at the end of each elongation step. The PCR mixture consisted of 10 µL SsoFast™ EvaGreen^®^ Supermix, 7 µL nuclease-free water, 1 µL cDNA, and 1 µL of each forward and reverse primer. Melting curve analysis was performed to ensure that single amplicons were obtained for each target gene after each qRT-PCR and primer efficiencies were determined using at least five different cDNA concentrations. Primers were designed on an exon-intron junction using the NCBI Primer tool [45]. Gene expression was evaluated using the ΔΔCt method [46,47]. The genes actin beta and glyceraldehyde-3-phosphate dehydrogenase (GAPDH) were used as endogenous references for normalizing target mRNA. Data are presented as means ± standard error (SE) from three biological replicates. Results were analysed using the GraphPad Prism 5 software (version 5.01; GraphPad Software, San Diego, CA, USA) and the significance was determined using the unpaired Student’s *t*-test. *p*-values < 0.05 were considered significant.

### 2.6. SDS-PAGE and Immunoblotting

MDA-MB-231 cells were grown in 35 mm Petri dish, incubated with PEVs for 24 h, and lysed with Lysis Buffer (50 mM Tris–HCl, 150 mM NaCl, 1% Nonidet P-40, 1 mM EDTA, 0.25% deoxycholic acid, 0.1% SDS, pH 7.4, 1 mM PMSF, 5 μg/mL leupeptin, and 5 μg/mL aprotinin). Protein expression and phosphorylation were analysed by SDS-PAGE followed by immunoblotting, as previously described [48]. Membrane staining was performed using the different antibodies diluted 1:1000 in TBS (20 mM Tris, 500 mM NaCl, pH 7.5) containing 0.5% BSA and 0.1% Tween-20 in combination with the appropriate HRP-conjugated secondary antibodies (1:2000 in PBS plus 0.1% Tween-20). Quantification of the band intensity was performed by computer-assisted densitometric scanning using Quantity One–4.6.8 software (BioRad).

### 2.7. Statistical Analysis

All the reported figures are representative of three different experiments. As to the Ca^2+^ data, the amplitude of Ca^2+^ release in response to extracellular stimulation was measured as the difference between the ratio at the peak of intracellular Ca^2+^ mobilization and the mean ratio of 1 min baseline before the peak. The magnitude of SOCE evoked by extracellular stimulation upon Ca^2+^ restoration to the bath was measured as the difference between the ratio at the peak of extracellular Ca^2+^ entry and the mean ratio of 1 in baseline before Ca^2+^ re-addition. The rate of Mn^2+^ influx was evaluated by measuring the slope of the fluorescence intensity curve at 400 s after Mn^2+^ addition [40,42]. Pooled data are given as mean ± SEM, while the number of cells analysed is indicated above the corresponding histogram bars (number of responding cells/total number of analysed cells).

For immunoblotting and cell migration analyses, all reported figures are representative of at least three different experiments and the quantitative data are reported as mean ± SD. Data from immunoblotting scanning were normalised considering the intensity of the band of interest in the sample not incubated with PEVs as one arbitrary unit (A. U.). For cell migration analyses, data are expressed as number of migrated cells per field at 10X microscope magnification.

Comparisons between two groups were done using the Student’s *t*-test, whereas multiple comparisons were performed using One-Way Analysis of Variance (ANOVA) with the Bonferroni post-hoc test. *p*-values less than 0.05 were considered statistically significant. Data were analysed using the GraphPad Prism (Version 5.01) software.

## 3. Results

### 3.1. Long-Term Exposure to PEVs Increases ER Ca^2+^ Mobilization and SOCE Activation in MDA-MB-231 Cells

The dose-response relationship showed in a previous study from our group demonstrated that 30 µg/mL represents the PEV dose that is more efficiently internalized and is more effective at inducing migration in MDA-MB-231 cells [16,49]. The long-term effect of PEVs on intracellular Ca^2+^ dynamics was therefore evaluated in MDA-MB-231 cells maintained or not (control, Ctrl) in the presence of 30 µg/mL PEVs for 24 h and loaded with the Ca^2+^-sensitive fluorophore, Fura-2/AM. Preliminary recordings revealed that PEVs did not affect either the basal [Ca^2+^]_i_ (Appendix A) or the modest constitutive Ca^2+^ entry that has previously been reported in MDA-MB-231 cells (Appendix A) [50]. Next, we exploited the Ca^2+^ add-back protocol to evaluate whether long-term exposure to PEVs influences ER Ca^2+^ mobilization and SOCE activation. As described elsewhere [27,28,30,37,50,51], this manoeuvre consists in stimulating the cells with cyclopiazonic acid (CPA), a selective inhibitor of SERCA, in the absence of extracellular Ca^2+^ (0Ca^2+^) to cause passive ER Ca^2+^ efflux through Ca^2+^-permeable leakage channels. Subsequently, 1.8 mM Ca^2+^ is restituted to external Ca^2+^ to monitor SOCE through Orai1 channels that have previously been activated by STIM1 upon the depletion of the ER Ca^2+^ store. Figure 1A,B show that CPA (10 µM) caused a transient increase in [Ca^2+^]_i_ under 0Ca^2+^ conditions, which reflects ER Ca^2+^ releasing ability in the absence (Ctrl) and presence of PEVs. PEVs caused a significant (*p* < 0.05) elevation in CPA-evoked ER Ca^2+^ mobilization (Figure 1C). The following restitution of extracellular Ca^2+^ induced a second increase in [Ca^2+^]_i_ (Figure 1A,B), which was entirely due to SOCE through Orai1 channels [27,28,30,50]. Statistical analysis revealed that long-term exposure to PEVs also increased SOCE in MDA-MB-231 cells (Figure 1D). Altogether, these findings indicate that PEVs induce a remarkable remodelling of the Ca^2+^ handling machinery in highly aggressive breast cancer MDA-MB-231 cells by enhancing both ER Ca^2+^ release and SOCE.

### 3.2. Long-Term Exposure to PEVs Increases InsP_3_-Induced ER Ca^2+^ Release and InsP_3_-Sensitive SOCE

The migratory mechanism in MDA-MB-231 cells exposed to chemotactic cues involves an initial phase of Ca^2+^ release from the ER through InsP_3_Rs followed by SOCE activation [26,30,31,52,53]. Adenosine triphosphate (ATP) represents an autocrine/paracrine messenger that has long been used to evaluate InsP_3_-dependent Ca^2+^ signals in multiple types of cancer cells [23,40,53,54], including MDA-MB-231 cells [55,56]. ATP binds to G_q_-coupled P_2Y_ receptors, thereby stimulating phospholipase Cβ (PLCβ) to cleave phosphatidylinositol-4,5-bisphosphate (PIP_2_) into diacylglycerol (DAG) and InsP_3_ [23,40,54,57]. Preliminary experiments confirmed that ATP-evoked intracellular Ca^2+^ release was mediated by InsP_3_Rs (Appendix A). The Ca^2+^ add-back protocol confirmed that both phases of the Ca^2+^ response to ATP (100 µM), i.e., intracellular Ca^2+^ release through InsP_3_Rs and SOCE, were significantly enhanced in MDA-MB-231 pre-treated with PEVs for 24 h when compared to untreated (Ctrl) cells (Figure 2A–C). As described elsewhere [37,40,57], the agonist was removed from the perfusate 100 s before the re-addition of extracellular Ca^2+^ to prevent the activation of second messengers-operated channels and ionotropic P_2X_ receptors. To circumvent the G_q_-protein-PLCβ-InsP_3_ signalling pathway, InsP_3_Rs were directly activated with a membrane-permeant esterified form of InsP_3_, known as InsP_3_-BM (1 µM) [58]. The Ca^2+^ add-back protocol showed that InsP_3_-BM-evoked ER Ca^2+^ mobilization and SOCE were significantly larger in MDA-MB-231 pre-treated with PEVs for 24 h when compared to untreated (Ctrl) cells. Overall, these findings provide further support to the notion that long-term exposure to PEVs potentiates InsP_3_-dependent Ca^2+^ signals in MDA-MB-231 cells.

### 3.3. Molecular Characterization of the Ca^2+^ Handling Machinery in MDA-MB-231 Cells Exposed to PEVs

In order to gain further insights into the molecular mechanisms whereby PEVs increase InsP_3_-induced ER Ca^2+^ mobilization and SOCE activity, we carried out a thorough RT-qPCR analysis of mRNAs isolated from MDA-MB-231 cells exposed or not (Ctrl) to PEVs (30 µg/mL; 24 h). Figure 3 shows a significant increase in the expression levels of the transcripts encoding for SERCA2B (Figure 3A), i.e., the major SERCA isoform in MDA-MB-231 cells [59], and for InsP_3_R1 (Figure 3B) and InsP_3_R2 (Figure 3C), which mediate ER Ca^2+^ release [26]. Conversely, there was no change in the expression of the transcripts encoding for InsP_3_R3 (Figure 3D) and for all the molecular components of the SOCE machinery in MDA-MB-231 cells, i.e., STIM1 (Figure 3E) and Orai1 (Figure 3F) [24,27,30,31]. Notably, RT-qPCR analysis of PEV transcripts did not reveal detectable levels of any of these mRNAs (data not shown), thereby showing that SERCA2B, InsP_3_R1, and InsP_3_R2 transcripts are not directly transferred to MDA-MB-231 cells from PEVs (mean Ct values ± SD of 37.5 ± 1, 38.5 ± 1, 38.2 ± 1.1, 34.9 ± 1.4, 34.8 ± 1, 37.3 ± 1.4, and 36.6 ± 1.3 for SERCA2B, InsP_3_R1, InsP_3_R3, Orai1, SERCA3, InsP_3_R2, and STIM1, respectively).

Next, western blot analysis of SERCA2B, InsP_3_Rs, STIM1, and Orai1 protein expression was performed by exploiting affinity-antibodies, as illustrated in [37,40,53]. Immunoblots revealed a major band of ≈115 kDa for SERCA2B (Figure 4A, left panel) and a large band over 250 kDa, deriving from the sum of the 313/260/250 kDa bands corresponding to InsP_3_R1, InsP_3_R2, and InsP_3_R3 (Figure 4B, left panel), as previously shown in [53]. Furthermore, two major bands of ≈86 and ≈35 kDa were detected for, respectively, STIM1 (Figure 4C, left panel) and Orai1 (Figure 4D, left panel). Densitometric analysis confirmed that SERCA2B (Figure 4A, right panel) and InsP_3_R (Figure 4B, right panel) proteins were significantly up-regulated in MDA-MB-231 cells exposed to PEVs (30 µg/mL; 24 h). Conversely, and in agreement with the qRT-PCR data, there was no significant difference in the expression level of STIM1 (Figure 4C, right panel) and Orai1 proteins (Figure 4D, right panel). Overall, these findings strongly suggest that the increase in the amount of ER Ca^2+^ that is releasable through InsP_3_Rs is due to the overexpression of SERCA2B and InsP_3_R proteins. Since there is no difference in the expression levels of its underlying constituents, i.e., STIM1 and Orai1, the increase in SOCE is likely to reflect the larger drop in ER Ca^2+^ concentration following agonist stimulation of MDA-MB-231 cells pre-exposed to PEVs [60,61,62].

### 3.4. Serum-Induced Intracellular Ca^2+^ Signals Are Larger in MDA-MB-231 Cells Exposed to PEVs

It has long been known that FBS stimulates the migration in cancer cells through an increase in [Ca^2+^]_i_ [26,30,31,52,53]. FBS consists of a mixture of growth factors that bind to specific tyrosine kinase receptors (TKRs) that are coupled to PLCγ and stimulate InsP_3_-production and InsP_3_-dependent Ca^2+^ signalling. A recent report from our group showed that 24 h treatment with PEVs potentiated serum-evoked migration in MDA-MB-231 cells [16]. Therefore, we reasoned that the increase in the chemotactic response to FBS could be associated to an elevation in the underlying increase in [Ca^2+^]_i_. The Ca^2+^ add-back protocol confirmed that, upon long-term exposure to PEVs (30 µg/mL; 24 h), MDA-MB-231 cells displayed a remarkable increase in serum-evoked intracellular Ca^2+^ release and SOCE activation when compared to non-treated (Ctrl) cells (Figure 5A–C). The increase in extracellular Ca^2+^ entry was further confirmed by measuring the area under the curve (AUC) corresponding to the increase in [Ca^2+^]_i_ evoked by the re-addition of extracellular Ca^2+^ (Appendix A). Pharmacological characterization revealed for the first time that serum-evoked intracellular Ca^2+^ release was mediated by ER Ca^2+^ release through InsP_3_Rs in MDA-MB-231 cells. Indeed, the intracellular Ca^2+^ response to 20% FBS was dramatically reduced by each of the following manoeuvres [16,37,40,63]: (1) blockade of PLCγ activity with the specific aminosteroid, U73122 (10 µM); (2) specific inhibition of InsP_3_Rs with 2-APB (50 µM) (Figure 5D,F); and (3) depletion of the ER Ca^2+^ store with CPA (30 µM) (Figure 5E,F).

In agreement with previous reports on other cancer cell types [40,52], serum-evoked SOCE (Figure 6A) was dampened by blocking Orai1 channels with either Pyr6 (10 µM) (Figure 6B,D) or BTP-2 (20 µM) (Figure 6C,D) in MDA-MB-231 cells. Taken together, these findings show that long-term exposure to PEVs leads to an increase in InsP_3_-induced ER Ca^2+^ mobilization and SOCE that could potentiate the pro-migratory Ca^2+^ response to serum stimulation.

### 3.5. PEVs Potentiate Serum-Dependent Migration through the Ca^2+^-Dependent Recruitment of p38 MAPK and MLC2 in MDA-MB-231 Cells

To assess whether the partial remodelling of the Ca^2+^ handling machinery potentiates serum-induced migration upon pretreatment with PEVs, we evaluated MDA-MB-231 motility in the absence (Ctrl) and presence of specific InsP_3_R and SOCE inhibitors. The pharmacological blockade of InsP_3_Rs with 2-APB (50 µM) significantly inhibited serum-migration in MDA-MB-231 cells pre-treated with PEVs (30 µg/mL, 24 h), whereas this process was not affected by SOCE inhibition with BTP-2 (20 µM) (Figure 7A,B). Next, we assessed whether a serum-evoked increase in [Ca^2+^]_i_ is required to boost p38 MAPK and MLC2 activation in the presence of PEVs (30 µg/mL, 24 h). We confirmed that PEVs potentiated p38 MAPK and MLC2 phosphorylation (Figure 7C–F), while they did not hyper-activate the Ca^2+^-sensitive Pyk2 (Appendix A). We then examined the phosphorylation of p38 MAPK and MLC2 in the absence (Ctrl) and presence of 2-APB and BTP-2. Unlike migration, PEV-dependent increase in p38 MAPK and MLC2 phosphorylation was sensitive both to InsP_3_R inhibition with 2-APB (50 µM) and to SOCE blockade with BTP-2 (20 µM) (Figure 7C–F). These findings demonstrate that InsP_3_Rs, rather than SOCE, support PEV-dependent migration in MDA-MB-231, although they can both induce p38 MAPK and MLC2 phosphorylation.

## 4. Discussion

Circulating platelets can contribute to cancer progression by releasing PEVs that facilitate (by shrouding disseminated tumour cells from recognition by NK cells) or stimulate cancer cell spreading from the primary site [6,13]. Multiple pieces of evidence support the notion that PEVs promote cancer cell invasiveness by mediating the transfer of platelet-derived instructive cues, consisting either in signalling proteins or genetic material, to the target cells [6,13,64]. The relationship between breast cancer and platelets is complex and is yet to be fully unravelled. Nevertheless, PEVs have also been shown to directly engage signal transduction pathways, such as Src, focal adhesion kinases (FAKs), p38 MAPK, and MLC2, which stimulate migration in the TNBC MDA-MB-231 cell line [16,65]. In accord, MDA-MB-231 cells can induce platelet activation and aggregation [49,66], thereby leading to the release of robust amounts of PEVs that, in turn, increase cancer cell migration and invasion [16]. Intriguingly, an increase in [Ca^2+^]_i_ may occur upstream of p38 MAPK and MLC2 recruitment [16,18] and can modulate p38 MAPK and MLC2 phosphorylation [67,68,69]. It has long been known that a complex remodelling of Ca^2+^ handling machinery supports several cancer hallmarks, including migration and invasion [21,22,70]. Therefore, we sought to investigate whether the long-term exposure of the highly invasive MDA-MB-231 cells to PEVs potentiates migration by rewiring their Ca^2+^ transport system.

### 4.1. PEVs Induce Remodelling of the Ca^2+^ Handling Machinery in MDA-MB-231 Cells

A sustained increase in [Ca^2+^]_i_ is required to trigger the release of PEVs from the plasma membrane of activated platelets [4]. Preliminary reports suggested that PEVs can, in turn, also influence intracellular Ca^2+^ dynamics in target cells. For instance, a recent investigation showed that PEVs elicit intracellular Ca^2+^ oscillations in aortic vascular smooth muscle cells, thereby promoting migration and neointimal hyperplasia in a rat model of vascular injury [71]. Furthermore, we documented that 30 µg/mL PEVs induced a robust increase in [Ca^2+^]_i_ in MDA-MB-231 cells, with this being initiated by InsP_3_-induced ER Ca^2+^ release and maintained by SOCE activation [16]. Nevertheless, MDA-MB-231 cells displayed enhanced migration and invasiveness upon 24 h incubation with PEVs [16]. Therefore, the immediate Ca^2+^ response to PEVs is unlikely to engage the Ca^2+^-dependent molecular machinery that potentiates migration during the late stages of PEV stimulation. These pieces of evidence prompted us to assess whether long-term exposure to PEVs stimulates the p38 MAPK and MLC2 signalling pathways and potentiates MDA-MB-231 cell migration through remodelling the Ca^2+^ handling machinery.

#### 4.1.1. PEV-Dependent Increase in InsP_3_-Induced ER Ca^2+^ Release Is Associated to SERCA2B and InsP_3_R Up-Regulation

Pre-treatment with PEVs did not change either the resting [Ca^2+^]_i_ or the basal plasma membrane permeability to Ca^2+^, which is mediated by a yet-to-be-identified Ca^2+^-permeable route in MDA-MB-231 cells, while it is mediated by Orai1 channels in low-migrating MCF-7 cells [41]. Since PEVs did not affect resting Ca^2+^ influx in MDA-MB-231 cells, we did not further investigate its molecular structure.

The Ca^2+^ response of breast cancer cells to chemotactic cues is triggered by InsP_3_-induced Ca^2+^ release from the ER and sustained over time by SOCE [24,26,32]. The Ca^2+^ add-back protocol represents the most common strategy to evaluate both the amount of ER Ca^2+^ that can be released through InsP_3_Rs and the extent of SOCE activation in cancer cells, including MDA-MB-231 cells [27,28,30,37,50]. We found that, in the absence of extracellular Ca^2+^, CPA-evoked intracellular Ca^2+^ release was significantly increased in MDA-MB-231 cells exposed to PEVs when compared to untreated cells. The amount of Ca^2+^ introduced into the cytosol via ER Ca^2+^ leakage channels upon SERCA inhibition with CPA or thapsigargin is recognized as a reliable indicator of the releasable ER Ca^2+^ pool. In accord, this protocol has been widely exploited to evaluate the differences in ER Ca^2+^ content in primary vs. metastatic cancer cells [40] and in cancer cells exposed to different pharmacological or genetic treatments [26,30,54,57,72,73]. Molecular analysis showed that long-term exposure to PEVs caused an increase in the transcript and protein levels of SERCA2B, which represents the most abundant SERCA isoform in MDA-MB-231 cells [59]. Therefore, the increase in SERCA2B expression might be responsible for the increase in the ER Ca^2+^ load unmasked by CPA-evoked intracellular Ca^2+^ mobilization. Similarly, the increase in SERCA2B expression underlies the larger Ca^2+^ releasing ability of colorectal cancer cells lacking the oncogenic K-Ras isoform, K-Ras^G13D^ [54]. The increase in ER Ca^2+^ content in PEV-treated MDA-MB-231 cells is associated to an increase in InsP_3_R1 and InsP_3_R2 transcript and protein expression. The Ca^2+^ add-back protocol confirmed that both physiological (with ATP) and pharmacological (with InsP_3_-BM) stimulation of InsP_3_Rs resulted in a significant elevation in InsP_3_-dependent ER Ca^2+^ mobilization in MDA-MB-231 cells exposed to PEVs. An increase in InsP_3_R1 expression may contribute to multiple oncological processes, including apoptosis resistance in prostate cancer [74], autophagy induction in clear cell renal cell carcinoma [75], and proliferation, invasion, and migration in osteosarcoma [76]. Similarly, InsP_3_R2 regulates migration in non-small cell lung cancer [77], maintains the self-renewal ability of liver cancer stem cells [78], and prevents apoptosis in B-cell lymphoma and chronic lymphocytic leukemia [79]. Interestingly, both InsP_3_R1 and InsP_3_R2 proteins drive migration in MDA-MB-231 cells [26], whereas InsP_3_R1 has also been shown to promote proliferation [80]. Therefore, the combined overexpression of SERCA2B and InsP_3_R1/InsP_3_R2 proteins nicely correlates with the increased ER Ca^2+^ content and higher level of InsP_3_-induced ER Ca^2+^ mobilization induced by long-term exposure to PEVs and could potentiate migration in MDA-MB-231 cells [16]. PEVs can transfer specific cargo molecules, such as mRNA, DNA, cytokines, and membrane receptors or enzymes, to recipient cells and thereby increase the capability of invasive behaviour in cancer cells [3,6,13]. Nonetheless, we could not find detectable levels of SERCA2B and InsP_3_R1/InsP_3_R2 transcripts in the mRNA content of PEVs. Therefore, we can conclude that long-term exposure to PEVs stimulates the partial remodelling of the Ca^2+^ handling machinery in MDA-MB-231 cells by boosting the expression of genes encoding for SERCA2B, InsP_3_R1, and InsP_3_R2.

#### 4.1.2. PEV-Induced SOCE Potentiation Does Not Involve STIM1 and ORA1 Up-Regulation

The larger ER Ca^2+^ release could also lead to an increase in SOCE activation even though the expression of its underlying molecular components, i.e., STIM1 and Orai1, is not altered by PEVs [60,61,62], as shown in the present investigation. Although some studies have provided evidence against a straightforward relationship between the extent of ER Ca^2+^ depletion and SOCE amplitude [81,82], other investigations have clearly shown a roughly linear relationship between the magnitude of InsP_3_-induced ER Ca^2+^ release and SOCE activation [60,61,62,83,84]. In agreement with these observations, SOCE amplitude in MDA-MB-231 cells treated with PEVs was always significantly larger than in untreated cells whatever the stimulus inducing ER Ca^2+^ release, i.e., CPA, ATP, or InsP_3_-BM. Interestingly, SOCE has also been shown to drive metastasis and invasion in MDA-MB-231 cells [24,30,31]. Therefore, the overall remodelling of intracellular Ca^2+^ dynamics, i.e., the enhancement of InsP_3_-induced ER Ca^2+^ release and SOCE activation, is predicted to stimulate migration in PEV-treated MDA-MB-231.

### 4.2. PEV-Dependent Increase in InsP_3_-Induced Ca^2+^ Release Potentiates Migration in MDA-MB-231 Cells: The Role of p38 MAPK and MLC2 Signalling Pathways

An increase in [Ca^2+^]_i_ has long been known to mediate the pro-migratory effect of serum on cancer cells [26,30,31,52,53]. In line with this evidence, early work showed that PLCγ1 is recruited by serum to stimulate motility and adhesion [85] and that genetic silencing of Orai1 prevents serum-induced migration in MDA-MB-231 cells [31]. Herein, we provided the first characterization of the molecular mechanisms whereby serum triggers an increase in [Ca^2+^]_i_ in this highly migrating breast cancer cell line. Pharmacological manipulation confirmed that serum-evoked intracellular Ca^2+^ release in MDA-MB-231 cells was inhibited by interfering with PLC activity with U73122 by inhibiting InsP_3_Rs with 2-APB and by depleting the ER Ca^2+^ pool with CPA. Furthermore, serum-evoked extracellular Ca^2+^ entry was dampened by BTP-2 and Pyr6, two highly specific inhibitors of Orai1. These findings concur with the involvement of InsP_3_Rs and SOCE in the Ca^2+^ signal evoked by serum in MDA-MB-231 cells, as also documented in other types of cancer cells [40,53,86], including low-migrating MCF-7 breast cancer cells [87]. As expected by the preliminary characterization with CPA, ATP, and InsP_3_-BM, both phases of serum-evoked intracellular Ca^2+^ signals (i.e., ER Ca^2+^ mobilization and SOCE) were significantly increased upon exposure to PEVs. Previous work has shown that both InsP_3_Rs and SOCE drive migration in highly migrating MDA-MB-231 breast cancer cells [24,26,32]. Therefore, the potentiation of serum-evoked intracellular Ca^2+^ signals could contribute to the enhanced migration rate that we recently reported in MDA-MB-231 cells exposed to PEVs [16]. In agreement with this hypothesis, blocking InsP_3_Rs with 2-APB significantly reduced serum-induced migration, a finding that is entirely consistent with the reported involvement InsP_3_R1 and InsP_3_R2 in MDA-MB-231 cell migration [26]. Surprisingly, inhibiting SOCE with BTP-2 did not affect motility in PEV-treated MDA-MB-231 cells. Nevertheless, both 2-APB and BTP-2 prevented serum-induced MLC2 and p38 MAPK phosphorylation, which drives cancer cell migration [17,18,19,20]. These data confirm our recent evidence that the long-term exposure to PEVs potentiates p38 MAPK and MLC2 phosphorylation in MDA-MB-231 cells [16] and further shows that the recruitment of these signalling pathways requires an increase in [Ca^2+^]_i_. Under the same conditions, we failed to detect any effect with regard to PEVs on the activation of the Ca^2+^-sensitive focal adhesion kinase Pyk2, whose role in driving the breast cancer metastatic outgrowth was previously documented [16]. The seeming discrepancy between the divergent effects of 2-APB and BTP-2 on p38 MAPK and MLC2 phosphorylation and cell motility could reflect some degree of redundancy between multiple Ca^2+^ sources which can engage the same Ca^2+^-dependent effectors [88,89,90]. However, the phosphorylation cascades triggered by SOCE do not seem to play a major role in MDA-MB-231 cell migration, which is instead finely tuned by the p38 MAPK and MLC2 pathways engaged by InsP_3_Rs. Alternatively, InsP_3_-induced ER Ca^2+^ release could recruit additional Ca^2+^-dependent effectors of cell motility that are not coupled to SOCE and remain to be identified. Future work is mandatory to assess this issue, but InsP_3_-induced ER Ca^2+^ release is clearly required to potentiate p38 MAPK and MLC2 phosphorylation and to stimulate migration in PEV-treated MDA-MB-231 cells.

## 5. Conclusions

Herein, we provide the first evidence that long-term exposure to PEVs induces a remarkable alteration in the Ca^2+^ transport system in the highly aggressive TNBC cell line, MDA-MB-231, thereby leading to an increase in InsP_3_-induced Ca^2+^ release and SOCE amplitude. In particular, the larger Ca^2+^ mobilization from the ER is required to potentiate serum-induced migration through the recruitment of p38 MAPK and MLC2. These findings lay the foundation for targeting the Ca^2+^ handling machinery not only to prevent breast cancer cell stimulation by pro-oncogenic chemical mediators and physical signals [23,24,29,37,55,56] but also by PEVs.

## Figures and Tables

**Figure 1 cells-11-03120-f001:**
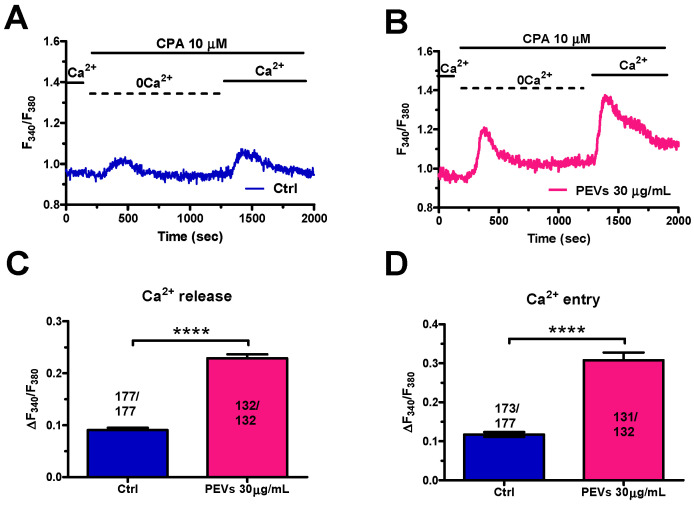
Long-term exposure of MDA-MB-231 cells to PEVs increases ER Ca^2+^ mobilization and SOCE activation. (**A**). Control (Ctrl) cells were challenged with CPA (10 μM), a selective SERCA inhibitor, in the absence of extracellular Ca^2+^ (0Ca^2+^). This manoeuvre caused a transient increase in [Ca^2+^]_i_ due to the depletion of the intracellular ER Ca^2+^ store. The restoration of extracellular Ca^2+^ induced a second increase in [Ca^2+^]_i_, which is representative of SOCE. (**B**). Cells treated for 24 h with PEVs (30 μg/mL) showed a higher Ca^2+^ transient when stimulated with CPA under 0Ca^2+^ conditions. Likewise, the restoration of extracellular Ca^2+^ induced an enhanced increase in [Ca^2+^]_i_ that is indicative of a greater SOCE. (**C**). Mean ± SEM of the CPA-dependent intracellular Ca^2+^ mobilization in control and PEV-treated cells. **** indicate *p* < 0.0001. (**D**). Mean ± SEM of the CPA-dependent SOCE amplitude in control and PEV-treated cells. **** indicate *p* < 0.0001.

**Figure 2 cells-11-03120-f002:**
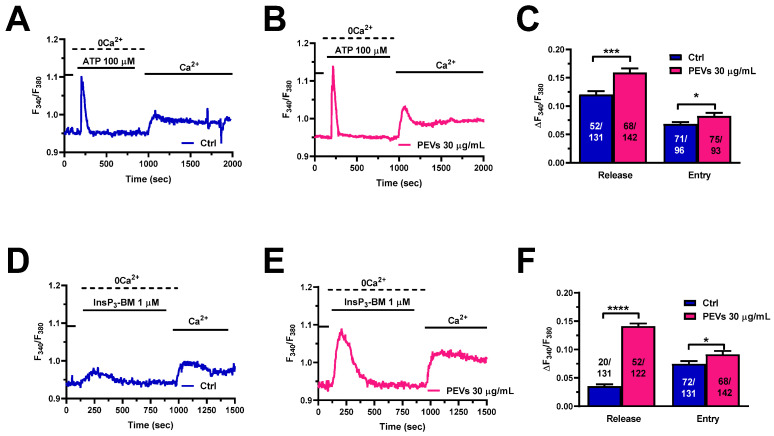
Long-term exposure of MDA-MB-231 cells to PEVs increases InsP_3_-induced ER Ca^2+^ mobilization and SOCE. (**A**). ATP (100 µM) evoked a transient increase in [Ca^2+^]_i_ under 0Ca^2+^ conditions in control (Ctrl) cells. ATP was then removed from the recording bath, while extracellular Ca^2+^ was restored to the solution to measure SOCE. (**B**). The treatment of MDA-MB-231 cells with PEVs (30 μg/mL, 24 h) increased ATP-induced endogenous Ca^2+^ mobilization and SOCE activation. (**C**). Mean ± SEM of ATP-induced intracellular Ca^2+^ release and SOCE in control (Ctrl) and PEV-treated cells. * indicates *p* < 0.05 and *** indicates *p* < 0.001. (**D**). InsP_3_-BM, a membrane-permeant esterified form of InsP_3_, induced a small increase in [Ca^2+^]_i_ under 0Ca^2+^ conditions in control (Ctrl) cells. Restoration of extracellular Ca^2+^, in the absence of the agonist, caused a second bump in [Ca^2+^]_i_ that was indicative of SOCE activation. (**E**). The treatment of MDA-MB-231 cells with PEVs (30 µg/mL, 24 h) caused an enhancement in intracellular Ca^2+^ mobilization and SOCE in MDA-MB-231 cells stimulated with InsP_3_-BM. (**F**). Mean ± SEM of InsP_3_-BM-induced endogenous Ca^2+^ mobilization and SOCE in control (Ctrl) and PEV-treated cells. **** indicates *p* < 0.0001 and * indicates *p* < 0.05.

**Figure 3 cells-11-03120-f003:**
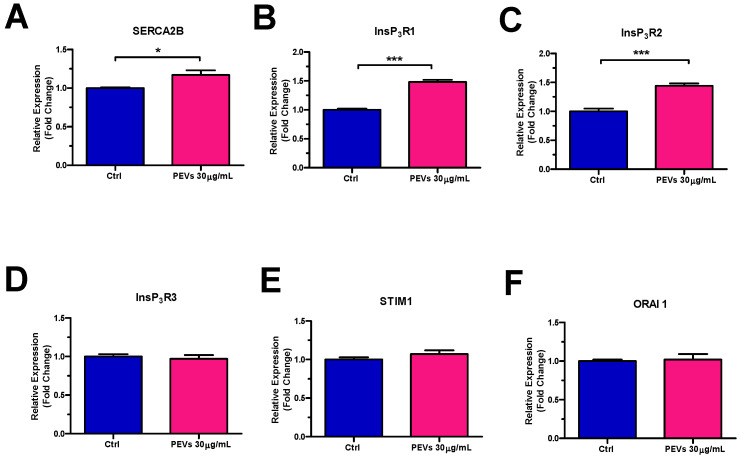
Long-term exposure to PEVs enhances the expression of SERCA2B, InsP_3_R1 and InsP_3_R2 transcripts. RT-qPCR was performed on the transcripts encoding for SERCA2B (**A**), InsP_3_R1 (**B**), InsP_3_R2 (**C**), InsP_3_R3 (**D**), STIM1 (**E**), and Orai1 (**F**) and harvested from MDA-MB-231 cells treated with 30 µg/mL for 24 h as compared to untreated (Ctrl) cells. Results were normalized with the level of the GAPDH control. * indicates *p* < 0.05, *** indicate *p* < 0.001.

**Figure 4 cells-11-03120-f004:**
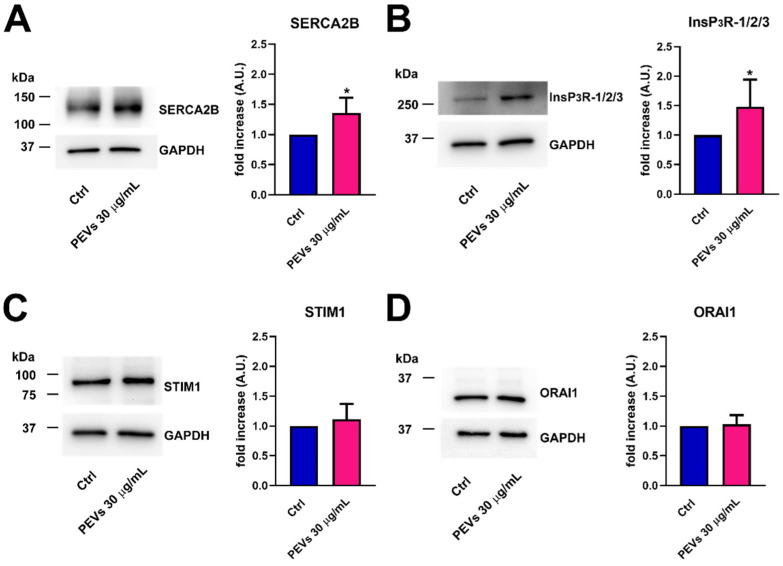
The treatment of MDA-MB-231 cells with PEVs increases the protein expression of SERCA2B and InsP_3_Rs. Representative immunoblots of SERCA2B and InsP_3_Rs are reported in (**A**,**B**), respectively, where GAPDH is for equal loading control (left panel). Quantification of the results performed by densitometric scanning is reported in the right panels as fold increase (A.U.) of protein expression over basal (control, Ctrl). Results are the mean ± SD of three different experiments. * indicates *p* < 0.05. Conversely, long term exposure of MDA-MB-231 cells to PEVs do not cause differences in the expression level of STIM1 and Orai1. The left panels of (**C**,**D**) show the representative immunoblots of STIM1 and Orai1, respectively, while the right panels show band quantifications reported as fold increase (A.U.) of protein expression over basal (Ctrl). Results are the mean ± SD of three different experiments.

**Figure 5 cells-11-03120-f005:**
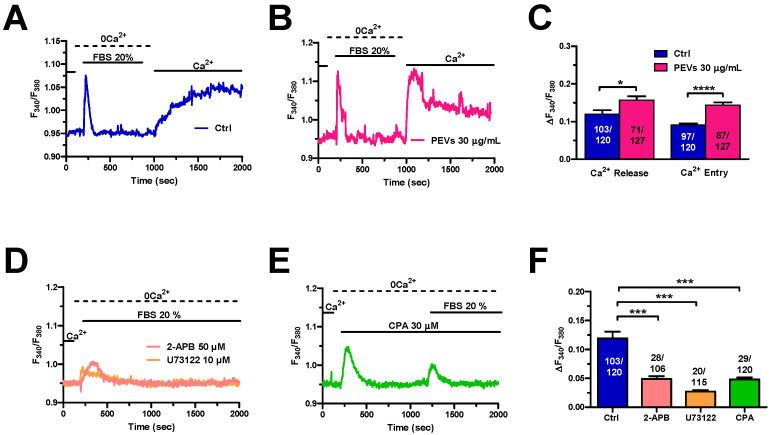
Long-term exposure to PEVs enhances FBS-induced ER Ca^2+^ mobilization and SOCE. (**A**). A transient increase in [Ca^2+^]_i_ was evoked by 20% FBS under 0Ca^2+^ conditions in control (Ctrl) cells. FBS was then removed from the recording bath, while extracellular Ca^2+^ was restored to the perfusate to measure SOCE. (**B**). Pre-treatment of MDA-MB-231 cells with PEVs (30 μg/mL, 24 h) increased the endogenous Ca^2+^ mobilization and the SOCE induced by 20% FBS. (**C**). Mean ± SEM of 20% FBS-induced intracellular Ca^2+^ release and SOCE in control and treated cells. **** indicate *p* < 0.0001 and * indicates *p* < 0.05. (**D**). The Ca^2+^ response to 20% FBS, under 0Ca^2+^ conditions, was inhibited by U73122 (10 µM, 30 min), a selective PLC blocker. Moreover, the Ca^2+^ signal was inhibited by blocking InsP_3_Rs with 2-APB (50 µM, 30 min). (**E**). Emptying the ER with CPA (30 µM, 20 min) prevented (not shown) or significantly reduced an FBS-induced increase in [Ca^2+^]_i_. (**F**)**.** Mean ± SEM of the amplitude of 20% FBS-evoked Ca^2+^ peak under the designated treatments. *** indicate *p* < 0.001.

**Figure 6 cells-11-03120-f006:**
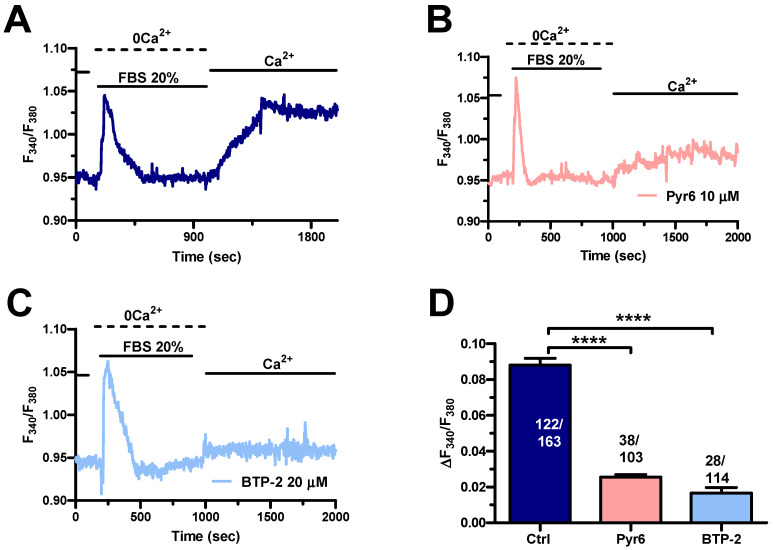
The Ca^2+^ response to FBS is sustained by SOCE in control MDA-MB-231 cells. (**A**). An intracellular Ca^2+^ response was induced by 20% FBS under 0Ca^2+^ conditions. Restoration of extracellular Ca^2+^, in the absence of the agonist, induced a second increase in [Ca^2+^]_i_, which was indicative of SOCE. (**B**). Extracellular Ca^2+^ entry evoked by 20% FBS was suppressed or significantly reduced by blocking Orai1 with Pyr6 (10 µM, 10 min). (**C**). Extracellular Ca^2+^ entry evoked by 20% FBS was suppressed or significantly reduced by blocking Orai1 with BTP-2 (20 µM, 10 min). (**D**). Mean ± SEM of the amplitude of Ca^2+^ peak in cells under the designated treatments. **** indicate *p* < 0.0001.

**Figure 7 cells-11-03120-f007:**
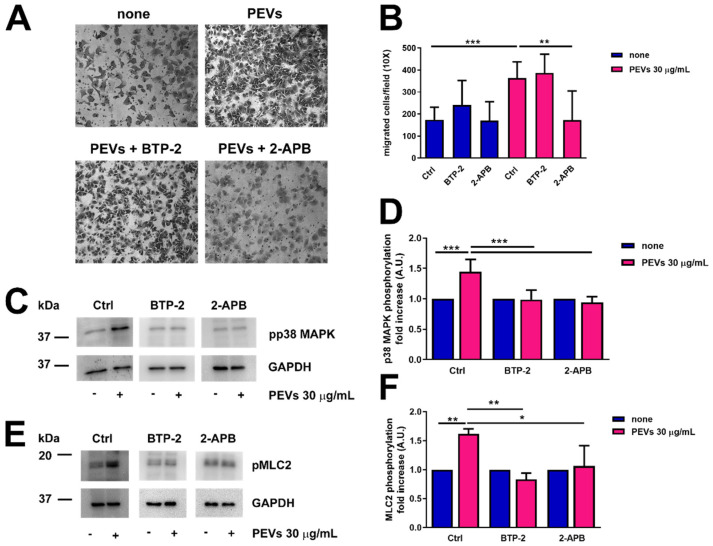
Pharmacological blockade of InsP_3_Rs with 2-APB significantly inhibited migration in MDA-MB-231 cells treated with PEVs. (**A**). MDA-MB-231 cells were incubated with PEVs (30 µg/mL, 24 h) or left untreated (none) before being treated with 2-APB (50 µM) and BTP-2 (20 µM) and then transferred into cell culture inserts. The cells migrated through the porous membrane were stained and counted. Representative images of migrated stained cells are reported. The quantification of the results is shown in (**B**) as the mean ± SD of three experiments. ** *p* < 0.01 and *** *p* < 0.005. Phosphorylation of p38MAPK (**C**,**D**) and MLC2 (**E**,**F**) in MDA-MB-231 cells incubated with PEVs (30 µg/mL, 24 h) and then treated with 2-APB (50 µM) and BTP-2 (20 µM). Representative immunoblots are reported in (**C**,**E**) for phospho-p38MAPK and phospho-MLC2, respectively, where GAPDH staining is for equal loading control. The quantification of the results is shown in (**D**) for phospho-p38MAPK and in (**F**) for phospho-MLC2 as the mean ± SD of three experiments. * indicates *p* < 0.05, ** indicate *p* < 0.01 and *** indicate *p* < 0.005.

## Data Availability

Data supporting reported results can be obtained upon reasonable request to the authors.

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
