# Peer review of "Platelet-Derived Extracellular Vesicles Stimulate Migration through Partial Remodelling of the Ca2+ Handling Machinery in MDA-MB-231 Breast Cancer Cells"

_cells, 2022, doi:10.3390/cells11193120_

Round 1
Reviewer 1 Report
This is a good study exploring the role of platelet derived extracellular vesicles (PEVs) in migration of MDA-MB-231 cells caused by in InsP3-induced Ca2+ release.
Strengths:
-The study is the first to provide evidence that long term exposure to PEVs increase ER Ca2+ mobilization with SOCE (store-operated Ca2+) activation and associating it with SERCA2B and InsP3R up-regulation in MDA-MB-231 cells.
-The experimental design is good.
-Figures are well explained and consistent with the text.
Minor comments:
-Source (healthy or cancer patients) of the human blood platelets to generate the PEVs is not mentioned.
Reviewer 2 Report
Major points
1. Introduction P3, L68, the authors should explain bioactive compounds in detail. In particular, which components (enzyme, genes, etc) are responsible for upregulated calcium signaling?
2. P5, L235, the authors had better show the dose-response relationship in the MDA-MB-231 cell line employed in the present study.
3. P6 L284, The reviewer cannot see any significant enhancement in SOCE when compared to the representative traces shown in Fig. 2A and 2B. Please clear it.
4. P8, L345, SERCA upregulation seems not so clear in the blot (Figure 4A). The authors should replace it with a clearer one.
5. P9, L387, In the control, the SOCE rose over time until 2000 s while in the presence of PEVs it peaked at about 1000 s and declined thereafter. The author should quantitate the difference by comparing the aria under the curve rather than the F340/F380 ratio.
Minor points.
1. Several typo errors. ex. P6, L275. trisphosphate (ATP).
2. Mix American and British English spelling. ex. P6, L277 signalling.
[鈴木1]Speling mistake
